Differential bicodon usage in lowly and highly abundant proteins

http://orcid.org/0000-0001-8052-4880 Diambra Luis A. ldiambra@gmail.com
Centro Regional de Estudios Genómicos, Universidad Nacional de La Plata, CONICET , La Plata , Argentina
Wilke Claus
Electronic publication date: 2017 Mar 9
Publication date: 2017
Volume: 5
Electronic Location ID: e3081
Received 2016 Nov 1; Accepted 2017 Feb 10
Copyright: © 2017 Diambra
Copyright year: 2017
Copyright holder: Diambra
License: This is an open access article distributed under the terms of the Creative Commons Attribution License, which permits unrestricted use, distribution, reproduction and adaptation in any medium and for any purpose provided that it is properly attributed. For attribution, the original author(s), title, publication source (PeerJ) and either DOI or URL of the article must be cited.
License URL: https://creativecommons.org/licenses/by/4.0/

Keywords: Synonymous codon usage, Codon pairs, Co-translational folding, Translational rate

Funding: CONICET, PIP #0020 This work was supported by CONICET, PIP: #0020. The funders had no role in study design, data collection and analysis, decision to publish, or preparation of the manuscript.

==============================
Degeneracy in the genetic code implies that different codons can encode the same amino acid. Usage preference of synonymous codons has been observed in all domains of life. There is much evidence suggesting that this bias has a major role on protein elongation rate, contributing to differential expression and to co-translational folding. In addition to codon usage bias, other preference variations have been observed such as codon pairs. In this paper, I report that codon pairs have significant different frequency usage for coding either lowly or highly abundant proteins. These usage preferences cannot be explained by the frequency usage of the single codons. The statistical analysis of coding sequences of nine organisms reveals that in many cases bicodon preferences are shared between related organisms. Furthermore, it is observed that misfolding in the drug-transport protein, encoded by MDR1 gene, is better explained by a big change in the pause propensity due to the synonymous bicodon variant, rather than by a relatively small change in codon usage. These findings suggest that codon pair usage can be a more powerful framework to understand translation elongation rate, protein folding efficiency, and to improve protocols to optimize heterologous gene expression.

Introduction

The central dogma of molecular biology establishes that the information that specifies which amino acid monomers will be added next during protein synthesis is coded in one or more nucleotide triplets known as codons (Watson et al., 2003). The genetic code establishes a set of rules that associate the 20 amino acids and a stop signal with 64 codons. This code is almost universal with a few exceptions (Jukes & Osawa, 1993). As there are more codons than encodable signals (amino acids and stop signal), the genetic code is considered degenerated. However, it is well known that synonymous codons are not used with the same frequency. Biased codon usage is a pervasive feature of the information encoded in genomes, but it is not universal because different species have different associated preferences (Watson et al., 2003). The existence of selective pressures to promote codon usage bias highlights the complex nature of synonymous codon choices (Hershberg & Petrov, 2008; Quax et al., 2015). Early reports have pointed out that the bias in prokaryotes is towards codons with high translation rates (Gouy & Gautier, 1982; Pan, Dutta & Das, 1998). In this sense, Guimaraes, Rocha & Arkin (2014) established that, in Escherichia coli, the elongation rate is affected by specific amino acid composition, as well as by codon bias. On the other hand, the impact of codon usage on translational rates in eukaryotes, where mRNA processing can also affect the overall translational rate, is an active topic of research (Tuller, Kupiec & Ruppin, 2007; Tuller et al., 2010; Vogel et al., 2010; Waldman et al., 2010; Camiolo, Farina & Porceddu, 2012; Pop et al., 2014). However, the role of codon usage has gone beyond translational rates, because new experimental findings suggest that codons with slow translation rates temporally separate the synthesis of defined protein portions and can synchronize the synthesis with the concurrently folding process of the protein domains (Lemm & Ross, 2002; Kimchi-Sarfaty et al., 2007; Zhang, Hubalewska & Ignatova, 2009; Buhr et al., 2016). It has been shown that translational pauses can schedule the sequential folding schemes, leading to different protein conformations (Buhr et al., 2016), and that the functionality of translated proteins can be affected by replacing rare codons with more frequently used codons (Komar, Lesnik & Reiss, 1999; Spencer et al., 2012; Bali & Bebok, 2015). In addition to the use of rare codons associated with scarce tRNA usage, other mechanisms exist that modulate the speed of translation or cause pauses. Among these, one can mention the blocking of ribosomal transit due to secondary structure elements in mRNAs (Nackley et al., 2006), and interactions of basic residues in the nascent polypeptides with the wall of the ribosomal exit tunnel (Gloge et al., 2014). However, in the last years emerging evidence has shown that translational rate could be encoded by a sequence longer than a triplet of nucleotides, in particular by bicodons (Guo et al., 2012). In this sense, a study encompassing 16 genomes has revealed that bicodons formed by two rare codons are frequently found in prokaryotes but rarely used in eukaryotes (Buchan, Aucott & Stansfield, 2006). More recently, Gamble et al. (2016) showed that some adjacent codon pairs modulate translational elongation in Saccharomyces cerevisiae. By studying the expression of 35,811 synthetic variants of a superfolder GFP, they identified 17 adjacent codon pairs (hereafter bicodons) associated with a substantial reduction of the translation elongation rate. The observed reduction in the tranlation efficiency could be consequence of the wobble interactions involved with these bicodons (Gamble et al., 2016). In addition, bicodons such as NNUANN are universally underrepresented, whereas others such as NNGCNN are mostly preferred (Tats, Tenson & Remm, 2008). It was also reported that rare arginine codons, followed by proline codons, were among the slowest translated bicodons (Chevance, Le Guyon & Hughes, 2014). This evidence could be consequence of the codon co-occurrence bias mechanism (Cannarozzi et al., 2010; Zhang et al., 2013), or the kinetics of the mRNA translocation from the A-site to the P-site (Khade & Joseph, 2011). Codon pair bias was also observed in several viral genomes, which matched the codon usage bias of the host (Wong et al., 2010). This fact has been used to produce synthetic viruses with attenuated virulence as a new strategy for vaccine development (Coleman et al., 2008).

Thus, coding sequences seem to carry more information than that strictly needed for specifying the linear sequence of amino acids in a protein. This additional information is linked with the overall synthesis rate of the associated protein, and the pauses required for it to acquire its correct native structure. Despite the enormous impact that this subliminal code has on biotechnology, there are a few systems biology methods to associate nucleotide sequences with the rate of protein synthesis (Pop et al., 2014). Among these, one can mention the sequencing of ribosome-protected mRNA fragment or ribosome profiling. This methodology has been used to correlate mRNA levels with codon decoding times (Dana & Tuller, 2014; Gamble et al., 2016; Mohammad et al., 2016). In this paper, I present an alternative manner to identify coding sequences that can modulate ribosomal transit on the mRNA. In this comprehensive survey, I performed a statistical analysis of bicodon usage frequencies over two sets of proteins: the low protein abundance (PA) set and the high PA set, for nine organisms. The main finding is that there is an important bias of bicodon usage depending on PA. In this sense, I determine which bicodons are statistically associated with low or high translational rates, and in which cases this bias can be explained or not by codon usage bias. I found that the bicodons reported in Gamble et al. (2016) are almost never used to encode highly abundant proteins in yeast.

Furthermore, I present suggesting evidence for the role of bicodons in encoding translational rate. In this sense, I found that the alteration in the structure and function of the MDR1 protein (Kimchi-Sarfaty et al., 2007) associated with a synonymous single polymorphism can be better explained by a relatively big change in a pause propensity score than by a moderate change in codon usage.

Materials and Methods

Data sources

In this work, I have used two kinds of data: (i) genome-wide PA across nine model organisms, and (ii) nucleotide sequences associated with the proteins indicated above. The absolute PA data from three prokaryotes (Microcystis aeruginosa, Bacillus subtilis, and E. coli), one unicellular fungus (S. cerevisiae), one plant (Arabidopsis thaliana), two multicellular eukaryotes (Drosophila melanogaster and Caenorhabditis elegans), and two mammals (Mus musculus and Homo sapiens) were downloaded from the PaxDb web site (http://pax-db.org/) on May 2015 (Wang et al., 2012). From these comprehensive data sets, I selected one sample of the most abundant proteins, and another sample of the less abundant ones. When more than one isoform was present in the comprehensive data set, only one was included in the samples. It is important to point out that PA correlates negatively with coding-sequence length in yeast (Coghlan & Wolfe, 2000). As PA distributions are generally biased, i.e., short proteins should be more abundant than larger ones (see Fig. 1), I selected two sets of 500 sequences, in which the sequence length distribution of both sets was similar. The procedure for sampling sequences with similar length distributions, consisted in ordering all the sequences which corresponded to a given organism in a PA increasing order. To select the sequences of the low PA samples, I began from the lowest PA extreme of the sequence list, and I compared rl=exp[−(l−lo)2/2σ] (where l is the length of the sequence, lo and σ are the mean length and standard deviation (SD), respectively, of the target distribution) with a random number uniformly distributed r. If rl > r, the sequence was added to the set of low PA sequences. Then, I tested the second sequence in a similar manner and so on, until 500 sequences were selected. To select the high PA sequence set, I performed the same procedure, but beginning from the highest PA extreme of the sequence list. The resulting lists of coding sequences corresponding to these unbiased samples are given in Tables S1 and S2. In Fig. 1 (and also in Figs. S1–S8), I have plotted the distributions of the whole PA for all organisms used in this study, and the PA and the sequence length distribution of the selected data sets. The nucleotide coding sequences corresponding to the selected proteins were downloaded from Ensembl web sites (four eukaryote organisms from ftp://ftp.ensembl.org/pub/ and four prokaryote organisms from http://bacteria.ensembl.org), while A. thaliana coding sequences were downloaded from www.arabidopsis.org.

Figure 1 Protein abundance and sequence length distributions.

Protein abundance distributions of the whole dataset of S. cerevisiae, low PA and high PA subsets are indicated in red and blue colors, respectively (A). Sequence length distributions of the subsets of sequences are shown in panel (B). Protein abundance distributions of the whole dataset, the selected low and high PA subsets of sequences used in the study (listed in Tables S1 and S2), are indicated in red and blue colors, respectively (C). Sequence length distributions corresponding to the subsets of sequences are shown in panel (D).

I also used the cumulative ribosome occupancy computed by Gamble et al. (2016) from the yeast ribosome profiling data set obtained by Jan, Williams & Weissman (2014). The cumulative ribosome occupancy is defined as the ratio between the sum of joint counts at positions with the bicodon in the ribosomal P-, A-sites and E-, P-sites and the joint counts sum across all surrounding window positions, which were extracted from Table S5 of Gamble et al. (2016).

Statistical analyses

Bicodon bias was studied in the context of the low and high PA samples. I counted all consecutive pairs of codons on the same reading frame of the coding sequences belonging to a given sample, which allowed us to compute the occurrence of each bicodon ij for all the sequences of each sample. The index i indicates the codon corresponding to P-site, while j indicates the one corresponding to the A-site. The occurrence of the codon pair ij will be denoted by oij. I also computed the number of single codons fi for all the sequences of each sample.

In summary, I analyzed the bias of bicodon usage in the two samples using three complementary measures: (i) the pause propensity score, which is based on the differential bicodon usage in both samples; (ii) the Fisher’s exact test, which establishes whether the bicodon usage bias is significant; and (iii) the residual score proposed in Gutman & Hatfield (1989), which establishes whether the bias in bicodons can be explained by the codon usage bias, or not.

The pause propensity score, denoted by π, is defined as the difference between the relative synonymous bicodon usage computed over the low PA sequences (RSBUL), and the one computed over the sequence sample associated with high PA (RSBUH). Mathematically, πij=RSBUijL−RSBUijH=qap(fijL−fijH)/Nap.(1)

Here, fijX is the frequency of the bicodon ij computed over the sequence sample X, qap is the number of bicodons encoding for the same amino acid pair, and Nap is the frequency of that amino acid pair for both samples. Thus, a large, or small, value of πij indicates the preference of bicodon ij for encoding low, or high, PA sequences, respectively. These values were clustered using a hierarchical average linkage over P-site and A-site codons according to patterns of similar bicodon preference.

Further, I use Fisher’s exact test to examine whether the number of occurrences of bicodon oijL, observed in the sample of sequences associated with lowly abundant proteins was significantly different to the number of occurrences observed in the sample of sequences associated with highly abundant proteins oijH. Thus, I constructed a 2 × 2 contingency table for each bicodon, as shown for an illustrative purposes in Fig. S9 for the particular case of bicodon AAGAAG (Agresti, 1992). In order to compute the p-value, I approximated the factorial operator with Stirling’s formula, n!≈2πn(n/e)n for n ≥ 25. To improve visualization, the colors in the heat maps are related to the quantity −Slog10(p-value) where S takes value +1 or −1, when the bicodon has preference for sequences with low or with high PA, respectively.

Furthermore, I computed the expected number of occurrences of each codon pair, as eij=fifjNp/Ntot2, where Ntot is the total number of codons in the set of sequences and Np is the number of bicodons. Following (Gutman & Hatfield, 1989), I removed the contribution due to the nonrandomness of amino acid pairs by normalizing the former expected values as:e^ij=eij×∑kl*Okl∑kl*ekl,(2)

where the * indicates that the sum is only over codon pairs kl encoding the same amino acid pair encoded by the bicodon ij. From the observed and normalized expected bicodon counts recorded in a given sample X, I computed the residual scores χXij for each codon pair as:χXij2=(oij−e^ij)2e^ij,(3)

where X indicates the sequence samples, i.e., X = L for low PA sample, or X = H for high PA sequence sample. These residual scores can be used to assess whether the bias in a given codon pair can be explained, or not, by the bias in codons and amino acids.

In order to statistically assess the residual scores, I performed a random shuffling control. From each sample of sequences, I generated a second random set of sequences by shuffling the order of codons (but preserving the stop codon at the end of sequence). This procedure removed the codon correlation but not the codon usage. Then, I computed the residual score of bicodons associated with this random sample. I repeated the above procedure 200 times and, finally for each bicodon ij I computed the mean value 〈χij2〉ran, and associated standard deviation SDran(χij2). Thus, the residual scores of bicodon ij can be expressed as the number of SDs from the mean, i.e., (χij2−〈χij2〉ran)/SDran(χij2). This procedure was performed for the low and high PA samples of sequences independently.

The above analysis was performed for all possible bicodons (61 × 64 = 3904), excluding stop:sense bicodons and the stop:stop bicodons. The values of frequencies, π, p-value, and χ2 for all bicodons and the nine organism are listed in Table S3. The scripts for each analysis in the manuscript are available in Code S1.

Results

Bicodon preferences

In the light of recent findings (Gamble et al., 2016), I am interested here in studying bicodons as determinants of translational elongation rate modulation in nine organisms. The working hypothesis was that sequences encoding abundant proteins should be optimized, in the sense of translation efficiency. This optimization could be reflected in the usage frequency of both codons and bicodons which encode sequences associated with high and low PA. Thus, I expected to capture this bias by statistical analysis of these sequence samples. To check this hypothesis, I selected a set of 500 coding sequences associated with highest abundance proteins, and another 500 coding sequences associated with lowest abundance proteins, in nine model organisms from different kingdoms. Before showing the whole analysis across several organisms, I begin with an illustrative example. Figure 2A shows the histogram of the bicodons (red bars) which encode for the amino acid pair VR, and which were obtained from 500 low PA sequences from S. cerevisiae.

Figure 2 Bicodon usage for the VR amino acid pair.

Red and orange bars denote the frequency of bicodons observed in a set of 500 coding sequences with low PA (A), and in a set of 500 coding sequences with high PA (B), respectively. Black bars represent the expected frequency obtained by the product of each codon frequency.

One can observe that bicodon usage is not uniform, i.e., it is biased, and this could be due to the known bias observed at the codon level. However, the expected frequency associated with such bicodons (black bars, obtained by the product of each codon frequency) shows that, although some bicodon frequencies can be explained by the bias in codon usage (for example, bicodon GTGAGA), other bicodons have an associated usage frequency that is underrepresented (such as bicodon GTTAGA), or overrepresented (as bicodon GTACGA). This means that two consecutive codons used to encode a given amino acid pair can be correlated. A similar analysis can be performed with sequences associated with high PA, as shown in Fig. 2B, and in all other amino acid pairs. Evidence for nonrandom associations between codon pairs, even once codon bias and bias against specific amino acid pairings were subtracted, was previously reported in E. coli (Gutman & Hatfield, 1989), and in many other genomes (Buchan, Aucott & Stansfield, 2006). However, what is a new remarkable fact in Fig. 2, is the strong difference between the histograms computed for the low PA and high PA samples. Fisher’s exact test allows one to reject, with a high significance level, the null hypothesis that bicodons are equally used in sequences from the low PA and high PA samples. In the particular case of bicodon GTTAGA, the p-value is 1.73 × 10−9. By comparison of the histograms corresponding to both low PA (red bars) and high PA (orange bars), it can be seen that some bicodons, such as GTTAGA or GTCCGT, are much more frequently used in sequences with high PA than in sequences with low PA, while the frequency usage of bicodon GTAAGA has an inverse relationship (see Fig. S10 for a direct comparison between both samples). More interestingly, the author found that some bicodons, such as GTACGG, GTACGA, or GTGCGA, are poorly or never used to encode VR amino acid pair in high PA sequences. In particular, the last two bicodons have been identified as inhibitors of translation elongation in yeast (Gamble et al., 2016). Figure 2 only illustrates the particular case of VR pair which is not a special case. There are many other bicodons with low frequency usage in high PA, and the 20 bicodons which mediate strong inhibition of translation elongation reported in Gamble et al. (2016) are a subset. Figure 3 depicts a raster plot of relative synonymous bicodon usage computed over the low PA sequences (RSBUL) vs. relative synonymous bicodon usage computed over the high PA sequences (RSBUH). The inset illustrates the region where inhibitor bicodons are. All of them, with the exception of CTTCTG, have an associated RSBUH value which is very small or zero. Besides these known inhibitor bicodons, many other bicodons exhibit usage bias between both samples, and these bicodons are far from diagonal line in the RSBUL–RSBUH plane (dashed line in Fig. 3). Bicodons with similar features could be expected in other organisms and, in order to assess the existence of this bicodon bias observed in S. cerevisiae in other organisms, I devised two alternative heat maps: one based on the pause propensity score, π, and another based on the p-value provided by the Fisher’s exact test.

Figure 3 Bicodon preference in S. cerevisiae.

The scatter plot of RSBUL vs. RSBUH illustrates the preference of bicodons in yeast. The inhibitor bicodons reported by Gamble et al. (2016) are indicated in red.

Pause propensity score across nine organisms

Figure 4 depicts a heat map for S. cerevisiae associated with the pause propensity score of 3904 bicodons. This score quantifies bicodon preference for coding low or high PA sequences. The position of each bicodon in the grid map was determined by clustering codons with similar pairing preference on both axes to facilitate the data visualization. Thus, bicodons with a clear preference for sequences associated with high PA (blue cells) have been grouped on the right-bottom corner of the grid, while red cells (high π score) indicate bicodons with preference for coding sequences associated with low PA. The author noticed that there are more bicodons with preference for encoding low PA sequences, than bicodons with preference for encoding high PA sequences, but the latter group has a stronger bias. Inhibitor bicodons identified in Gamble et al. (2016) are indicated by asterisks. All of them are associated with positive π scores, but not the highest π scores, which reflects the fact that these bicodons also have a smaller frequency associated with low PA sequences than other bicodons with preference for low PA. Another interesting feature of the heat map depicted in Fig. 4, is that some bicodons have an opposite preference depending on the order of the codons. For example, this is the case, to mention a few, for bicodons AGA-TGT and AGA-GCA (indicated by black circles), which have a preference for encoding low PA sequences (yellow cell), while the reverse ordered codon pairs TGT-AGA and GCA-AGA, respectively, have a preference for encoding high PA sequences (light blue cell). This fact cannot be explained solely in terms of codon usage bias.

Figure 4 Pause propensity heat maps for S. cerevisiae.

Each cell corresponds to the bicodon composed by the P-site codon (horizontal axis) and the A-site codon (vertical axis). Cell color is determined by the pause propensity score π associated with the bicodon. Red cells indicate bicodons with a clear preference for sequence associated with low PA, while blue cells indicate bicodons with high PA preference. Bicodons were grouped according to π similarity using average linkage clustering on each axis. Asterisks indicates the inhibitor bicodons identified in yeast (Gamble et al., 2016).

I also elaborated similar heat maps to the one depicted in Fig. 4, for the other eight organisms which are shown in Fig. 5. For the sake of comparison, the position of each cell has been preserved over all organisms. These maps reveal that discrepancy in bicodon usage between low and high PA samples is not the same across the studied organisms, but is a particular feature of each organism, the same as codon bias. However, C. elegans and S. cerevisiae have many bicodons with the same preference, a feature that is also shared to a lesser extent by D. melanogaster and A. thaliana. These similarities and discrepancies are more apparent when clustering bicodons for all nine organisms (see Fig. S11). Regarding the inhibitor bicodons identified in yeast (indicated by an asterisk in Fig. 5), one can see that many of them share a preference for low PA sequences with other organisms, in particular in the cases of C. elegans, A. thaliana, and D. melanogaster. On the other hand, both studied mammals (H. sapiens and Mus musculus), and E. coli, have less apparent preferences. Interestingly, it was in E. coli where bicodon bias was first reported (Gutman & Hatfield, 1989). However, it is clear that bicodon usage bias across the ORFeome does not imply a different usage preference between highly and lowly expressed proteins. The low preference of bicodon usage observed in E. coli can be a consequence of the fact that, in this particular case, there is not a clear distinction between PA distributions for both samples (see Fig. S6).

Figure 5 Pause propensity heat maps for all nine organisms.

Each cell corresponds to the bicodon composed by the P-site codon (horizontal axis) and the A-site codon (vertical axis). The position of bicodons in all the maps is the same as in Fig. 4. The color of the cell is determined by the pause propensity score π associated with the bicodon. Red cells indicate bicodons with a clear preference for sequences associated with low PA, while blue cells indicate bicodons with high PA preference. Bicodons were grouped according to π similarity using average linkage clustering on each axis. Asterisks indicate the inhibitor bicodons identified in yeast (Gamble et al., 2016).

The above heat maps show that bicodon usage bias exists in lowly and highly abundant proteins, but they do not indicate whether this bias is significant. For this, I applied Fisher’s exact test to compute the p-value against the null hypothesis of equal distribution in both low and high PA samples. Figure 6 depicts the corresponding heat map associated with S. cerevisiae, where the color of each cell in the grid is determined by the quantity −Slog10(p-value), where S takes the values +1 or −1 when the bicodon has preference for sequences with low or with high PA, respectively. This means that red cells indicate bicodons with significant bias towards low PA sequences, while blue cells indicate bicodons with significant bias to high PA sequences. Again, to facilitate data visualization, the position of each bicodon in the map was determined by clustering codons with similar pairing preference on the P-site and A-site axes. Inhibitor bicodons are indicated by asterisks. Some of them (like bicodons CGAGCG, CTCCCG, and CGACGG) are associated with a not significant bias (p-value < 0.01). This is due to the fact that these bicodons have very small frequency in low PA sequences and almost zero in high PA sequences. Something similar occurs for the sense:stop bicodons, which are grouped in three almost white rows at the center of the plot. For the sake of comparison, I also elaborated similar heat maps for other organisms where the positions of each cell has been preserved over all organisms (Fig. S12). A couple of white rows can be observed in all organisms. These cells correspond to bicodons which do not exhibit any preference or they are usually poorly used in both sequence samples and have poor associated statistics.

Figure 6 Fisher’s exact test for S. cerevisiae.

The color of each cell is determined by the quantity −Slog(p-value), where p-value is that provided by the Fisher’s exact test, S takes the values +1 or −1 when the bicodon has preference for sequences with low or with high PA, respectively. Thus, red cells indicate bicodons with a clear preference for sequences associated with low PA, while blue cells indicate bicodons with high PA preference.

The ribosome profiling method has been used for obtaining valuable information about protein synthesis, such as the location and strength of translational attenuation. Thus, sites associated with high levels of ribosome occupancy can be linked to translational pauses (Ingolia et al., 2009; Ingolia, Lareau & Weissman, 2011). To determine if ribosome occupancy correlates with the proposed pause propensity score, I used the cumulative ribosome occupancy for each bicodon, which was computed using the yeast ribosome profiling data (Table S5 of Gamble et al., 2016). Figure 7 depicts a scatter plot of the cumulative ribosome occupancy vs. the pause propensity score. Despite the fact that a clear correlation between these descriptors is not apparent in these plots, one can see that in addition to the inhibitor bicodons identified in Gamble et al. (2016), indicated by red circles, there are some other bicodons which have high levels of cumulative ribosome occupancy (> 0.03) and a significant preference for encoding low PA sequences (π > 0.75 and p-value < 0.01). These bicodons could be also candidates for modulating transcript elongation rate in yeast (they have been indicated by ** in Table S3).

Figure 7 Cumulative ribosome occupancy vs. π score and p-value for S. cerevisiae.

(A) Scatter plot of the cumulative ribosome occupancy and the pause propensity scores. (B) Scatter plot of the cumulative ribosome occupancy and quantity −Slog10(p-value). Red circles correspond to the known inhibitor bicodons. Other candidates for modulating transcript elongation rate are indicated in Table S3 by ** (π > 0.75, p-value < 0.01 and cumulative ribosome occupancy >0.03.

Codon bias cannot explain bicodon bias

The heat maps shown above are very useful to see some common features among organisms, but they do not show whether bicodon preference is explained or not by the preference of the codon in the pair for sequences associated with low or high PA. In order to study this, I have computed residual scores for each bicodon over sequences with low PA, χL2, and over sequences with high PA, χH2. When the residual score is high, bicodon usage cannot be explained by codon usage in the same sample of sequences, as it has been previously established in Gutman & Hatfield (1989) and Buchan, Aucott & Stansfield (2006). The value of these residuals scores for all codon pairs and organisms are listed in Table S3. In Fig. 8, a raster plot of these residual scores can be seen for all bicodons in B. subtilis, yeast, humans and E. coli (residual plots associated with the other five organisms are displayed in Fig. S13). As there are two sequence samples for each bicodon, it is convenient to take the quantity χ2=χL2+χH2 as a whole residual score. For all organisms, I have considered that those bicodons with χ ≥ 3 SD above the mean are bicodons whose preference for encoding low or high PA sequences is not explained by the preference of the individual codons in the pair (at a significance level of 0.003).

Figure 8 Bicodon bias cannot be explained by codon bias.

Scatter plots indicating the residual scores χL2 and χH2 computed over low PA and high PA samples, respectively, for B. subtilis, S. cerevisiae, H. sapiens, and E. coli. The axes units are the number of standard deviations from the meansobtained as indicated in methods. Raster plots of other organisms are displayed in Fig. S13. The codon pairs whose preference for sequences with low or high PA cannot be explained by codon usage bias are outside the grey quadrant (i.e., χ2 ≥ 3 SD above the mean). Among these, it is possible to distinguish bicodons more frequently used in low PA sequences (red squares), or in high PA sequences (blue triangles). Inside the quadrant, there are codon pairs with asignificantly different usage frequency in low and high PA samples, but whose bias can be explained by codon usage bias (green disks). Codon pairs whose usage frequencies in low and high PA samples are not significantly different aer indicated with black dots.

Thus, four types of bicodons can be distinguished in Fig. 8: (i) codon pairs that are significantly more used in sequences associated with low PA than in sequences associated with high PA (p-value < 0.01), and whose preferences cannot be explained by the codon usage bias (red dots); (ii) codon pairs which are significantly more used in sequences associated with high PA than in sequences associated with low PA (p-value < 0.01), and whose preferences cannot be explained by codon usage bias (blue dots); (iii) codon pairs with a significantly different usage frequency in low and high PA samples, but whose preferences can be explained by codon usage bias, i.e., χ2 < 3 × SD (green dots) and, finally, (iv) codon pairs whose usage frequencies in low and high PA samples are not significantly different, i.e., p-value > 0.01 (black dots).

These plots indicate that while there are many bicodons with evident preference for low and high PA sequences in C. elegans and S. cerevisiae, there are only few in H. sapiens at this significance level. Humans, mouse and, surprisingly, also E. coli, have few bicodons with evident preference. However, even in these organisms, there are numerous bicodons with similar usage frequencies in both samples, but whose associated frequency usage cannot be explained by codon usage bias (black dots out of the gray quadrant). This fact reveals that, even in these organisms, there is a complex correlation between two consecutive codons.

On the other hand, S. cerevisiae has more codon pairs than B. subtilis, with a significant different usage frequency in low and high PA samples. Nevertheless, the bias of many bicodons is explained by codon usage bias (green dots in S. cerevisiae panel).

Regarding the inhibitor bicodons, I found that only five of them have a significantly different usage frequency in low PA sample and only one in high PA sample (see Table S3). I searched for bicodons with the same preferences shared by S. cerevisiae, C. elegans, and D. melanogaster, by selecting those with p-value < 0.01 and χ2 ≥ 3 × SD above the mean. Table 1 lists 16 shared bicodons with preference for low PA, while Table 2 lists 40 bicodons with preference for high PA sequences. It is noteworthy that, even within each species, the number of bicodons with preference for encode high PA sequences (blue cells in Fig. 5) are smaller than the number of bicodons with preference for encoding low PA sequences (yellow cells in Fig. 5), and the number of shared bicodons in Table 2 is substantially greater than those listed in Table 1. This fact seems to indicate that bicodons with preference for high PA sequences are more conserved across these organisms than bicodons with preference for low PA sequences.

Table 1 Bicodons with preference for low PA.

Shared bicodons in C. elegans, D. melanogaster, and S. cerevisiae that have high preference for low PA (p-value < 0.01), but for which this preference cannot be explained by codon usage bias (χ2 > 3 SD above the mean).

Dipeptide	Bicodon	χL2 + χH2	−log10(p-value)	
Elegans	Fly	Yeast	Elegans	Fly	Yeast	
NA	AATGCA	3.57	21.46	4.88	7.06	4.57	4.07	
EL	GAACTA	3.20	3.23	4.58	2.45	6.25	2.30	
QK	CAGAAA	27.33	10.55	9.09	3.02	2.66	9.20	
SK	AGTAAG	19.71	51.08	10.35	2.31	4.40	2.46	
YK	TATAAA	12.21	6.43	4.01	4.00	5.23	6.05	
IK	ATTAAA	41.66	3.60	25.18	7.26	3.01	2.22	
AF	GCATTT	32.70	11.19	3.30	12.62	2.51	7.86	
FF	TTTTTT	3.52	8.28	4.598	13.53	3.72	12.19	
YP	TATCCG	8.87	69.04	15.73	2.54	3.38	7.40	
FQ	TTTCAG	7.52	4.82	3.79	7.90	5.29	6.46	
QS	CAAAGT	10.62	10.78	9.88	2.79	2.57	3.42	
ES	GAAAGT	9.73	7.50	33.32	5.48	2.58	5.31	
IG	ATAGGT	9.29	6.09	4.42	2.39	2.20	5.02	
NW	AATTGG	3.87	25.44	6.00	5.25	3.66	2.93	
KI	AAAATA	5.13	24.35	7.88	21.12	4.69	26.20	
SV	AGTGTG	25.90	11.45	3.66	2.16	2.98	3.78	

Table 2 Bicodons with preference for high PA.

Shared bicodons in C. elegans, D. melanogaster, and S. cerevisiae that have high preference for high PA sequences (p-value < 0.01), but for which this preference cannot be explained by codon usage bias (χ2 > 3 SD above the mean).

Dipeptide	Bicodon	χL2 + χH2	−log10(p-value)	
Elegans	Fly	Yeast	Elegans	Fly	Yeast	
AA	GCCGCC	11.85	40.65	3.17	7.60	7.90	2.37	
DA	GATGCT	15.28	9.96	16.66	2.13	2.03	2.25	
TA	ACCGCC	12.25	7.80	4.02	7.41	3.13	5.70	
IA	ATTGCC	3.05	143.23	35.23	4.45	9.06	4.18	
VA	GTTGCC	10.63	5.47	10.16	5.26	3.15	6.78	
RR	CGTCGT	3.62	4.25	34.28	11.91	7.12	8.11	
AK	GCCAAG	56.94	232.59	81.31	30.33	12.85	7.94	
NK	AACAAG	76.90	53.88	5.93	17.40	17.28	28.86	
DK	GACAAG	33.90	144.15	25.88	11.51	12.82	11.80	
FK	TTCAAG	34.07	14.43	39.10	18.98	4.90	10.80	
TK	ACCAAG	8.56	51.21	30.44	17.016	13.45	14.38	
YK	TACAAG	26.80	38.75	17.08	10.78	9.69	10.39	
IK	ATCAAG	92.63	73.15	31.05	27.48	14.95	22.05	
VK	GTCAAG	22.94	71.67	27.28	18.17	4.83	16.28	
AN	GCCAAC	48.52	34.31	5.87	13.30	11.83	3.34	
NN	AACAAC	19.27	52.78	9.09	5.03	6.21	15.14	
DN	GACAAC	35.91	77.21	15.15	5.41	10.01	4.75	
FN	TTCAAC	25.22	6.25	11.16	9.31	4.63	13.52	
TN	ACCAAC	29.56	26.29	13.69	9.64	8.03	15.67	
YN	TACAAC	25.17	26.30	8.26	2.74	3.23	8.62	
IN	ATCAAC	58.97	51.28	21.56	11.88	5.47	15.06	
TF	ACCTTC	3.71	10.08	3.77	7.68	6.84	3.09	
PP	CCACCA	6.17	60.99	3.47	16.27	3.65	3.65	
KS	AAGTCC	27.85	6.26	5.37	5.84	4.31	8.17	
AT	GCCACC	24.28	102.57	4.98	9.58	10.90	4.08	
NT	AACACC	14.60	6.294	5.81	7.15	5.43	12.79	
DT	GACACC	11.14	48.32	5.38	6.27	6.27	7.15	
FT	TTCACC	13.13	36.00	14.68	18.63	6.05	6.92	
ST	TCCACC	5.21	22.41	3.85	6.89	5.70	5.38	
TT	ACCACC	5.94	56.13	8.62	11.92	5.32	10.75	
IT	ATCACC	19.35	25.86	8.31	18.03	8.72	3.96	
VT	GTCACC	30.70	49.61	4.60	8.83	2.92	4.72	
DY	GACTAC	11.84	62.43	4.73	6.56	8.66	4.94	
AI	GCCATC	5.91	33.39	29.66	17.33	9.26	3.84	
FI	TTCATC	5.64	12.57	9.23	12.91	4.21	5.65	
TI	ACCATT	3.62	60.64	32.32	3.02	2.57	10.12	
TI	ACCATC	4.43	22.69	8.51	15.17	6.35	10.03	
II	ATCATC	12.55	27.02	14.52	27.31	8.41	12.78	
VI	GTCATC	15.21	71.68	5.70	11.29	2.76	8.11	
GV	GGTGTC	10.47	11.41	13.41	2.18	2.36	13.04	

Pause propensity score and cotranslational protein folding

The above statistical analysis is able to determine which bicodons are associated with lowly or highly abundant proteins. I hypothesized that bicodons associated with lowly abundant proteins could have a key role in programming translation pauses of the ribosomal machinery. As a proof of principle, I have identified one example that illustrates how even small biases as those observed in H. sapiens, could explain documented protein misfolding and a pathological condition in humans. It was previously reported that the single polymorphism (SNP) rs1045642 in the gene MDR1, which encodes the ABCB1 drug-transport protein, alters the structure and function of the protein with the consequent change in its substrate specificity (Kimchi-Sarfaty et al., 2007).

The SNP in exon 26 at position 3435 changes codon ATC to the synonymous ATT, which reduces codon usage from 47 to 35%. It was argued that the presence of a rare codon affects the timing of co-translational folding and insertion of P-glycoprotein into the membrane. Although it is difficult to consider the ATT codon as rare, it is clear that the SNP alters the timing of ribosomal transit. Here, I offer an alternative cause for translational attenuation. In this sense, I have observed that this SNP is also associated with a large change in the bicodon pause propensity score π. Specifically, bicodon ATCGTG has preference for low PA sequence with π = 0.1, while bicodon ATTGTG has preference for high PA sequence π = −0.8. This means a large change in comparison with the change in codon usage. Further, the other synonymous bicodon ATAGTG has a low pause propensity score, π = −0.55, in agreement with Kimchi-Sarfaty et al. (2007) observation that associates this haplotype to a larger decrease in the inhibitory effect. In addition to the SNP mentioned above, there are other synonymous SNPs related to human diseases that could be explained by a large change in the pause propensity. Among these, one can mention SNP rs34533956 in the gene CFHR5, which is associated with age-related macular degeneration (Narendra, Pauer & Hagstrom, 2009). In this case, the mutation changes bicodon GACGTG to GATGTG, and the associated change in pause propensity is from 0.03 to −0.78, while the change in the relative synonymous codon usage is only 13%. Another example corresponds to SNP rs11615 in the gene ERCC1, which was associated with colorectal cancer (Liang et al., 2010), where the pause propensity change is also large, against a small change in the relative synonymous codon usage. These relationships suggest that some pathological synonymous mutations could be understood in terms of the change in timing needed for co-translational folding programmed by bicodons.

Discussion

Organisms use a small fraction of the number of options offered by the genetic code redundancy. This is due to several constraints operating to optimize many important biological features such as expression level (Guimaraes, Rocha & Arkin, 2014), translational accuracy (Zaher & Green, 2009), protein solubility (Vasquez et al., 2016), folding accuracy (Sauna & Kimchi-Sarfaty, 2011; Spencer et al., 2012; Bali & Bebok, 2015), and protein stability, among others. In this sense, it has been shown that codon usage in E. coli is biased to reduce the cost of translational errors (Stoletzki & Eyre-Walker, 2007). In addition, codons that bind to their cognate tRNA more rapidly are used preferentially in highly expressed genes (Curran & Yarus, 1989). In this sense, Lian et al. (2016) have identified codons that regulate translation speed in human cell lines. A bias has also been reported in bicodon usage frequency in several organisms (Buchan, Aucott & Stansfield, 2006). More recently, Gamble et al. (2016) identified 17 bicodons associated with reduced translation efficiency from over 35,000 GFP variants in which three consecutive codons were randomized. Analysis of ribosome profiling data suggested that this translation modulation also operates in endogenous genes (Gamble et al., 2016). Many studies agree on the key role of ribosomal pause, in orchestrating the hierarchical co-translational folding of single domains (Kimchi-Sarfaty et al., 2007; Zhang, Hubalewska & Ignatova, 2009; Fluman et al., 2014; O’Brien, Vendruscolo & Dobson, 2014; Sander, Chaney & Clark, 2014; Tanaka, Hori & Takada, 2015; Buhr et al., 2016). In summary, there is increasing evidence that many relevant features, other than the linear sequence of amino acids, are also coded at nucleotide sequence level. These features should considerably reduce the amount of alternative ways to correctly convey the message from genes to functional proteins, despite the redundancy of the genetic code.

Among the above biological constraints determining codon usage, I have focused the attention on translational speed, i.e., the sequential process of protein elongation (Quax et al., 2015). Briefly, each iterative step of this process involves recruitment of the tRNA charged anticodons, tRNA association/dissociation to mRNA, assembly of the residue to the nascent peptide, and translocation of tRNA–mRNA from the A-site to the P-site. Each step has a particular rate, and it has been shown that disruption of the interaction between the mRNA codon in the A-site from the decoding center is a rate-limiting process (Khade & Joseph, 2011). In fact, there is evidence that such rates are codon dependent in E. coli (Gardin et al., 2014). Further contributions to the translational rate, not linked to the tRNAs’ abundance, are the non-Watson–Crick (wobble) interactions. These interactions are usually associated with higher dissociation rates between the mRNA and the decoding center (Spencer & Barral, 2012).

Even though the results provided here suggest that some bicodons could regulate translational attenuation, it is important to remark on the limitations of the present approach in assigning each bicodon with one value of the pause propensity index. The more evident limitation is that PAs are not uniquely dictated by a quick translation, since transcription levels also have important roles in prokaryotes (Guimaraes, Rocha & Arkin, 2014). Thus, it is possible that the observed bicodon bias could not only be due to an intrinsic translation efficiency, but also to the bias introduced by sequences associated with low or high transcript levels present in the sequence samples. To be able to dissociate the effect of transcript abundance in this present analysis, it would be necessary to be known the transcript abundance of the sequence in the sample in order to be able to normalize PA. On the other hand, the RNAi pathway is a common way to regulate expression in mammalians (Friedman et al., 2009). This factor could introduce an undesired bias and overshadow some bicodon bias. It is likely that the reduced number of bicodons with an evident preference which was observed in mammals, is due to the fact that PA is not majorly determined by the bicodon usage, with the consequent poor performance of the π score in these organisms. However, the residual score seems to be more robust analysis. Alternative methods based on ribosome density profile (Hussmann et al., 2015; Gamble et al., 2016; Mohammad et al., 2016), together with statistical analysis as the one proposed here, could overcome these drawbacks.

Conclusion

In this paper, I considered that ribosomal pauses are encoded by bicodons, and examined the bicodon frequency usage in nine organisms. I found that some codons have an evident preferential usage in sequences that code for highly abundant proteins, while many others have preference for encoding proteins that are scarce. The latter bicodons can be understood as short sequences linked to translational pauses. The observed bias cannot be explained by codon usage in many bicodons.

It is worth noting that few bicodons with differential bicodon usage in low and high PA sequences were found in E. coli, which clearly contrasts with the other prokaryotes studied here. For example, I reported almost 450 bicodons in B. subtilis which are preferentially used to encode either low or high abundant proteins without a codon usage correlation. Bicodon preference is also found in a plant, a fungus and two invertebrates. These results indicate that many bicodon preferences are shared by S. cerevisiae, C. elegans, and D. melanogaster and, to a lesser extent by A. thaliana and B. subtilis. Regarding the inhibitor bicodons identified in yeast, I found that they have a preference for encoding low PA sequence in agreement with Gamble et al. (2016). However, I found that there are additional bicodons associated with a high pause propensity score and high ribosome occupancy which deserve to be considered for further wet lab essays.

In the case of the mammalian species (H. sapiens and Mus musculus), I found a small number of bicodons with a clear preference. However, the number of bicodons whose frequency usage is not explained by codon usage is comparable to other organisms. To illustrate, I show an example of synonymous mutations of clinical relevance, in which the exchange of two codons with opposite preferences, even when such preferences are moderate, can alter translation ribosomal traffic. This example suggests that single mutations that change bicodon preference, can trigger pathological phenotypes by altering the translational attenuation program of the protein.

In summary, I report here that bicodon usage frequency depends on PA. This preference cannot be explained by the traditional codon usage for many bicodons. This empirical evidence supports the hypothesis that bicodons encode translation pauses. Such a scenario allowed us to contrast this hypothesis in various circumstances where translation rates could be altered. Like the naive codon usage, the bicodon usage can empower novel strategies for rational transcript design that minimize misfolding, while simultaneously maximizing co-translational folding for foreign proteins in heterologous hosts.

Supplemental Information

Supplemental Information 1 Unbiased sample of lowly abundant proteins for each organism.

List of 500 sequences used in our statistical calculations, first column is an internal index, second column corresponds to a gene, transcript or protein ID as provided by PaxDb database, third column is the protein abundance provided by PaxDb database, and the last column is the coding sequence length.

Click here for additional data file.

Supplemental Information 2 Unbiased sample of highly abundant proteins for each organism.

List of 500 sequences used in our statistical calculations, first column is an internal index, second column corresponds with a gene, transcript or protein ID as provided by PaxDb database, third column is the protein abundance provided by PaxDb database, and the last column is the coding sequence length.

Click here for additional data file.

Supplemental Information 3 Some features of 3,904 bicodons in the nine organisms.

First column corresponds to amino acid pair, second column to the corresponding bicodon, number of occurrences of such bicodon in the low (third column) and high (fourth column) PA samples. Fifth column corresponds to the divergence measure between histograms obtained from low and high PA samples of the dipeptide. The sixth and seven columns correspond to the residual scores χL and χH computed over the low PA and high PA samples, respectively. Pause propensities values of each bicodon are in the last column.

Click here for additional data file.

Supplemental Information 4 Protein abundance and sequence length distributions for A. thaliana.

The protein abundance distributions of the whole dataset, lowest and highest protein abundance subsets are indicated in red and blue colors, respectively (A). The sequence length distributions of the subsets of sequences are shown in the left panel (B). The protein abundance distributions of the whole dataset, and the selected low and high protein abundance subsets of sequences used in the study are indicated in red and blue colors, respectively (C). The sequence length distributions corresponding to the subsets of sequences are shown in the left panel (D).

Click here for additional data file.

Supplemental Information 5 Protein abundance and sequence length distributions for M. musculus.

The protein abundance distributions of the whole dataset, lowest and highest protein abundance subsets are indicated in red and blue colors, respectively (A). The sequence length distributions of the subsets of sequences are shown in the left panel (B). The protein abundance distributions of the whole dataset, and the selected low and high protein abundance subsets of sequences used in the study are indicated in red and blue colors, respectively (C). The sequence length distributions corresponding to the subsets of sequences are shown in the left panel (D).

Click here for additional data file.

Supplemental Information 6 Protein abundance and sequence length distributions for H. sapiens.

The protein abundance distributions of the whole dataset, lowest and highest protein abundance subsets are indicated in red and blue colors, respectively (A). The sequence length distributions of the subsets of sequences are shown in the left panel (B). The protein abundance distributions of the whole dataset, and the selected low and high protein abundance subsets of sequences used in the study are indicated in red and blue colors, respectively (C). The sequence length distributions corresponding to the subsets of sequences are shown in the left panel (D).

Click here for additional data file.

Supplemental Information 7 Protein abundance and sequence length distributions for C. elegans.

The protein abundance distributions of the whole dataset, lowest and highest protein abundance subsets are indicated in red and blue colors, respectively (A). The sequence length distributions of the subsets of sequences are shown in the left panel (B). The protein abundance distributions of the whole dataset, and the selected low and high protein abundance subsets of sequences used in the study are indicated in red and blue colors, respectively (C). The sequence length distributions corresponding to the subsets of sequences are shown in the left panel (D).

Click here for additional data file.

Supplemental Information 8 Protein abundance and sequence length distributions for D. melanogaster.

The protein abundance distributions of the whole dataset, lowest and highest protein abundance subsets are indicated in red and blue colors, respectively (A). The sequence length distributions of the subsets of sequences are shown in the left panel (B). The protein abundance distributions of the whole dataset, and the selected low and high protein abundance subsets of sequences used in the study are indicated in red and blue colors, respectively (C). The sequence length distributions corresponding to the subsets of sequences are shown in the left panel (D).

Click here for additional data file.

Supplemental Information 9 Protein abundance and sequence length distributions for E. coli.

The protein abundance distributions of the whole dataset, lowest and highest protein abundance subsets are indicated in red and blue colors, respectively (A). The sequence length distributions of the subsets of sequences are shown in the left panel (B). The protein abundance distributions of the whole dataset, and the selected low and high protein abundance subsets of sequences used in the study are indicated in red and blue colors, respectively (C). The sequence length distributions corresponding to the subsets of sequences are shown in the left panel (D).

Click here for additional data file.

Supplemental Information 10 Protein abundance and sequence length distributions for B. subtilis.

The protein abundance distributions of the whole dataset, lowest and highest protein abundance subsets are indicated in red and blue colors, respectively (A). The sequence length distributions of the subsets of sequences are shown in the left panel (B). The protein abundance distributions of the whole dataset, and the selected low and high protein abundance subsets of sequences used in the study are indicated in red and blue colors, respectively (C). The sequence length distributions corresponding to the subsets of sequences are shown in the left panel (D).

Click here for additional data file.

Supplemental Information 11 Protein abundance and sequence length distributions for M. aeruginosa.

The protein abundance distributions of the whole dataset, lowest and highest protein abundance subsets are indicated in red and blue colors, respectively (A). The sequence length distributions of the subsets of sequences are shown in the left panel (B). The protein abundance distributions of the whole dataset, and the selected low and high protein abundance subsets of sequences used in the study are indicated in red and blue colors, respectively (C). The sequence length distributions corresponding to the subsets of sequences are shown in the left panel (D).

Click here for additional data file.

Supplemental Information 12 Construction of contingency tables and Fisher’s exact test.

In the left table we show the observed occurrence for the 4 bicodons that encode the amino acid pair KK in low and high PA sequences samples of S. cerevisiae. From this table we can obtain one 2 × 2 contingency table for each bicodon as illustrated in the right table for the bicodon AAGAAG. In this particular case, the probability of obtaining such set of values (Fisher’s exact test) is around 5.3 × 10−93, consequently, we can reject the null hypothesis that this bicodon is equally likely to be in low and high PA samples. The formula to compute the p-value from the 2 × 2 contingency table is given in the bottom panel.

Click here for additional data file.

Supplemental Information 13 Frequency distributions from the unbiased samples of sequences.

Frequencies associated with bicodons that encode the amino acid pair VR, computed using sequences from low PA sample (red bars) and from high PA sample (orange bars). The frequency usage of bicodon GTGAGA in the sequences of both samples are almost the same, while other bicodons have an evident preference for sequences associated with low or high PA.

Click here for additional data file.

Supplemental Information 14 Pause propensity heat maps for nine organisms.

Cell color is determined by the pause propensity score associated to each bicodon. The matrix of 3,904 (bicodons) × 9 (organisms) were grouped according to the pause propensity score similarity using average linkage clustering over the nine organisms. Then, for a better visualization, we display the pause propensity scores of each organism in a 61 × 64 grid, preserving the position of bicodons on the grid across the organisms.

Click here for additional data file.

Supplemental Information 15 Heat maps associated to the Fisher’s exact test for nine organisms.

Cell color is determined by the quantity −Slog10(p-value), where p-value is provided by the Fisher’s exact test of the associated bicodon, S takes the values +1 or −1 when the bicodon has preference for sequences with low or with high PA. Thus, red cells indicate bicodons with clear preference for sequence associated to low PA, while blue cells indicate bicodons with high PA preference.

Click here for additional data file.

Supplemental Information 16 Residual plots associated to all studied organisms.

Scatter plots indicating the residual scores χL and χH computed over low PA and high PA samples, respectively. The codon pairs whose preference for sequences with low or high PA cannot be explained for codon usage bias are outside the grey quadrant (i.e., χ2 > 3 × SD). Among them, we distinguish bicodons more frequently used in low PA sequences (red dots), or in high PA sequences (blue dots). Inside the quadrant, there are codon pairs with a significantly different usage frequency in low and high PA samples, but such bias can be explained for codon usage bias (green dots). Codon pairs whose usage frequencies in low and high PA samples are not significantly different (black dots).

Click here for additional data file.

Supplemental Information 17 Source code.

Script code to perform all calculations on the manuscript in the Mathematica environment (Wolfram Inc.).

Click here for additional data file.

I thank Alejandra Carrea and Christina McCarthy for critical reading of the manuscript. Luis A. Diambra is member of CONICET (Argentina).

Additional Information and Declarations

Competing Interests

Author Contributions

Data Deposition

The authors declare that they have no competing interests.

Luis A. Diambra conceived and designed the experiments, performed the experiments, analyzed the data, contributed reagents/materials/analysis tools, wrote the paper, prepared figures and/or tables, and reviewed drafts of the paper.

The following information was supplied regarding data availability:

The raw data has been supplied as Supplemental Dataset Files.

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
