# Peer review of "Differential bicodon usage in lowly and highly abundant proteins"

_PeerJ, doi:10.7717/peerj.3081_

## Round 0.1 · original submission · Major Revisions

Both reviewers are generally positive about your work but point out some major issues that should be addressed.

Reviewer 1 ·

Basic reporting

The author presents a statistical analysis of bicodon usage in several organisms to infer a role of adjacent codon pairs in modulating translation speed. The main result is that in several cases bicodon preference is shared between related organisms. The work generally contributes to the growing understanding of the determinants of codon usage bias and translation kinetics. While the current manuscript is not of sufficient quality for publication, it should be reconsidered if the concerns listed below are addressed.

First and foremost, a recent beautiful paper experimentally characterized the role of adjacent codons in regulating translation elongation through screening of a large synthetic library of GFP variants in S.cerevisiae (Gamble et al. Cell 2016; PMID: 27374328). I would strongly recommend to the author to cite and build upon this work early on, e.g. by presenting a clean statistical analysis that generalizes and expands the findings from the work by Gamble et al. across additional organisms.

• The manuscript contains too many colloquial phrases. A thorough revision of the language used to increase clarity and professional/scientific English will greatly improve this manuscript

• The introduction, background, and referenced literature are generally adequate. The manuscript structure conforms to PeerJ standard.

• In the current form, not all figures are relevant, high quality, well labelled and described:

Figure 2: The y-axis labels could be bigger, and the y-axis of panel A should end with a tick mark and label to indicate the range of the plot. Relabeling the legend in the plot to ‘expected’ and ‘observed’ may clarify the figure, but this is merely a friendly suggestion. Moreover, the figure legend states that shown is “the frequency of bicodons observed in a set of 500 coding sequences”. I do not think that 30% (0.3) of codon duplets in 500 coding sequences are ‘AAAAAA’. The author should be careful to describe exactly and correctly what is shown (i.e. here the relative frequency of synonymous codon pairs)

Figure 3: The author should fix the y-axis. This is the same data as Figure 2 and a little superfluous. The author could add this graph as additional panel to Figure 2 for easier direct comparison.

Figure 4: This figure is not very informative in its current form. The figure legend does not specify which ‘statistical distance’ is used and illustrated. The coloring might be biased by extreme values in the top two panels as there seems to be almost no signal in the bottom two panels. The author could highlight codon pairs that have been found to significantly deviate from the expected frequency through the statistical analyses of this work to make this figure more informative.

Figure 5 is missing axes labels and units. I do not understand what is plotted here, thus cannot draw any conclusion from this figure.

• The data underlying the presented statistical analyses are provided in three Excel tables. Table1 and 2 are missing column names or any other form of annotations, so it is not clear what the numbers mean.

Experimental design

• Generally speaking, the study comprises original primary research within Scope of PeerJ.

• The research question is somewhat clear until the start of the results section, where the authors states: “The aim of this paper is to associate coding sequences with their relative translational speeds”, which differs markedly from the initial introduction as analysis of bicodon usage across organisms. The author should check that all statements are consistent and true to what was actually done.

• It is not clear why binomial coefficients are computed, as I cannot find further reference to them. Please clarify or remove them from the methods section

• Similarly, Shannon’s entropy is introduced in the methods section, but not referenced again. The paragraph describing Shannon’s entropy and the Kullback-Leibler divergence should be clarified and the exact wording carefully checked. A normalization through division by log(n) should be better justified.

• The statistical approach following (Gutman and Hatfield, 1989) is sound, appropriate, and sufficiently rigorous. The usefulness of Entropy and KL divergence to quantify differential bicodon usage is not necessarily demonstrated here. An alternative way to estimate how robust any statistical differences are would be e.g. an analysis of randomized sequences.

• Residual scores reported as statistically significantly deviating (between observed and expected) may be more informative than a classification based on an arbitrary threshold of 5.

• The author keeps making reference to p-values > 2 or more. This is against the definition of p-values and does not make sense. I assume the author means the negative log-transformed p-value, but this has to be clarified in the manuscript (and called something other than "p-value").

• The author also includes ribosome profiling data in his work, but does not specify at all how these data were processed and analyzed. The methods should be clarified to provide sufficient information to replicate all analyses.

Validity of the findings

• Data (genomic nucleotide sequences and curated expression datasets) is generally robust, and the approach statistically sound, and controlled. However, anti-Shine-Dalgarno sequences do not cause ribosome attenuation. The original paper (Li et al. Nature 2012), while a clever idea, has since been shown through careful biochemical work to be an artifact of sequencing and analyses methods applied (Mohammad et al. Cell Rep 2016; PMID: 26776510). Thus, some analyses by the author are based on an incorrect hypothesis. These have to be removed and/or corrected. I suggest, as mentioned above, to exploit the direct experimental evidence of a role of codon pairs in translation speed to establish a link between bicodon usage bias in multiple organisms with a likely function in regulating translation

• Generally, conclusions are well enough stated but speculation should be contained. For example the sentence “But first it is needed to solve the link between density and ribosomal speed at the nucleotide level, due to the fact that a pause at a given site will stop the transit of many other proofreading upstream ribosomes, increasing artificially the ribosome density of the upstream sequences. “ is not only grammatically incorrect, but also wrongly states that a translational pause necessarily will stop following ribosomes (if the transcript is translated by polysomes at all), as well as that ribosomes are ‘proofreading’. Similarly, the sentence “If we consider an average of three alternative codons for coding each amino acid, there exist more than 1.3×10143 manners to codify a protein with 300 residues.” needs to be corrected at multiple levels. I cannot follow this back-of-the-envelope calculation (and it would not be difficult to use the actual degeneracy of the genetic code to make this a correct statement instead of using an average degeneracy); this calculation seems to ignore any constraints on the amino acid sequence; ‘encode’ may be a better (more correct) word than ‘codify’; etc. There are many sentences of this sort in the present manuscript. All sentences throughout the manuscript should be carefully corrected for clarity and correctness.

Additional comments

The overall rationale of this work, is sound, but I strongly recommend that the author thoroughly revises the present manuscript to improve on clarify, presentation and correctness of all statements.

Reviewer 2 ·

Basic reporting

Structurally, the manuscript is laid out well, the introduction provides the appropriate context, and the literature is well referenced. The figures are also described clearly and raw data is easily accessible.

The text contains several grammatical errors, and inconsistent punctuation and capitalization (e.g. in gene names). These errors in punctuation and capitalization are also found throughout the References. Several paper titles in References are also missing words. The manuscript should be carefully proofread and these errors should be corrected.

Experimental design

The author presents a method of identifying coding sequences that may affect ribosomal movement along the mRNA by analyzing bicodons. The investigation is statistically sound and the author uses three different quantities (chi-squared, pause propensity, and Shannon's entropy) to support their conclusions about bicodon preferences in high and low abundant proteins.

In Methods, there are several mistakes that would make it difficult to reproduce the analyses in the manuscript:

- Line 101: I believe that this is the equation for a normal distribution, but it is missing a normalization factor.

- Equation 1: Variables k and l are not defined.

- Chi is used in several places when I think the author means chi-squared.

Validity of the findings

I have one major comment about the analyses presented in the manuscript. As the author mentions in Discussion, mRNA abundance could be a major confounding factor in these analyses. However, the Discussion and Conclusion are written under the assumption that different bicodons affect translation rather than transcription. The author should elaborate on how transcription and transcript abundances could confound analyses of bicodons and protein abundance. Additionally, it would be worthwhile to do an analysis of bicodon preferences in high and low PA sequences in E. coli, controlling for transcript abundances. Here is a data set that could be used that contains a matched set of protein and mRNA abundances:

Houser JR, et al. Controlled Measurement and Comparative Analysis of Cellular Components in E. coli Reveals Broad Regulatory Changes in Response to Glucose Starvation. PLOS Comp Biol 11(8): e1004400.
http://dx.doi.org/10.1371/journal.pcbi.1004400

As a minor comment, there are several typographical errors throughout the manuscript that, if interpreted at face-value, make the conclusions non-sensical. For example, in the Table 1 caption, the author states "(p-value > 3)" when I think they mean "(-log(p-value) > 3)". Again, the manuscript needs to be carefully proofread.

---

## Round 0.2 · Minor Revisions

Both reviewers were mostly satisfied with your revisions but had a few remaining minor requests. I'd like you to address those before I formally accept the paper. In particular, I agree with Reviewer #2 that source code should always be provided in a machine-readable format.

Reviewer 1 ·

Basic reporting

see below

Experimental design

see below

Validity of the findings

see below

Additional comments

The author has sufficiently addressed all prior comments and suggestions for corrections, which has clearly improve the manuscript. I would like to say that the manuscript now conforms to PeerJ standard and that I support publication in the current form. However, the following very minor issues should still be addressed:

The author should check formula (1). If q is a constant, it could be explicitly defined or dropped. Else, it might have to be q_ap?

I appreciate this sense of humor, but the author may want to consider whether to call out Fisher’s exact test as ‘celebrated’ (page 4, line 128) [I am just waiting for a student to come and ask about the difference between 'Fisher's exact test' and the 'celebrated Fisher's exact test' ...]

Two more typos: ‘thata’ should be ‘that’ (page 6, line 241), and ‘tas’ should be ‘as’ (legend of Fig. 5)

The main comment here is that I would recommend to the author to find a solution for Fig. 5 (and S11/S12) if possible. While it is stated that the codons follow the same order as in the previous corresponding figures, this is not immediately clear, which makes these figures hard to read. Maybe a large and shared axis across all panels with codon order, and possibly even the corresponding amino acid (even colored based on e.g. hydrophobic/polar/charged)? I’ll leave this to the author.

Reviewer 2 ·

Basic reporting

No comment

Experimental design

No comment

Validity of the findings

No comment

Additional comments

The author has addressed all of my concerns, except those regarding source code. While the author has provided source code in a PDF format, it would be more accessible as a raw text file. Please submit all code as a text file.

---

## Round 0.3 · accepted · Accept

Thank you for addressing the remaining reviewer comments.